# The epidemiology of autoimmune bullous diseases in Sudan between 2000 and 2016

**Omayma Siddig**[1]*, **Mayson B. Mustafa**[1], **Yousif Kordofani**[2], **John Gibson**[3], **Ahmed M. Suleiman**[1]

**1** Faculty of Dentistry, University of Khartoum, Khartoum, Khartoum State, Sudan, **2** Khartoum Dermatological and Venereal Diseases Teaching Hospital, Khartoum, Khartoum State, Sudan, **3** Institute of Dentistry, University of Aberdeen, Aberdeen, Aberdeenshire, The United Kingdom

* omayma.siddig@gmail.com

## Abstract

### Objectives

Autoimmune bullous diseases vary in their clinico-epidemiological features and burden across populations. Data about these diseases was lacking in Sudan. We aimed to describe the epidemiological profile and to estimate the burden of autoimmune bullous diseases in Sudan.

### Methods

This was a retrospective cross-sectional study conducted at Khartoum Dermatological and Venereal Diseases Teaching Hospital. We used routinely collected health care data, and included all patients with an autoimmune bullous disease who presented to the hospital between 2001 and 2016.

### Results

Out of the 4736 patients who were admitted to the hospital during the study period, 923 (19.5%) had an autoimmune bullous disease. The average rate of patients at the hospital was 57.7 per year representing 1.3 per 100,000 population per year. After exclusion of patients where the final diagnosis was missing, 585 were included in the further analysis. Pemphigus vulgaris was the most common disease (50.9%), followed by bullous pemphigoid (28.2%), linear IgA disease/chronic bullous disease of childhood (8.4%), and pemphigus foliaceous (8.2%). Pemphigoid gestationis and IgA pemphigus constituted 1.4% and 1.2% of the cohort, respectively. Paraneoplastic pemphigus, mucous membrane pemphigoid, lichen planus pemphigoidis, bullous systemic lupus erythematosus, and dermatitis herpetiformis were rare. None of the patients had epidermolysis bullosa acquisita.

### Conclusions

The clinico-epidemiological characteristics vary among the types of autoimmune bullous diseases. Females were more predominant in most of them. Sudanese patients tended in general to present at a younger age than other populations. The pool of Sudanese patients

**Data Availability Statement:** Data cannot be shared publicly because the local ethical regulations prohibit public sharing of patients' data. Data are available from the Faculty of Dentistry University of Khartoum Ethics Committee (contact

via e-mail:dentistry@uofk.edu) for researchers who
meet the criteria for access to confidential data.

**Funding:** The authors received no specific funding
for this work.

**Competing interests:** The authors have declared
that no competing interests exist.

with autoimmune bullous diseases is large which requires investigation for the local risk factors and presents a field for future trials.

## Introduction

Autoimmune bullous diseases (AIBDs) are a heterogenous group of mucocutaneous diseases that are characterized by the presence of circulating and tissue-bound autoantibodies targeting epidermal and dermal adhesion molecules [1]. They manifest clinically as bullae that rupture forming ulcerations. They run a chronic course and cause significant morbidity and mortality. AIBDs are classified according to the level of blistering into intraepidermal and subepidermal groups [2]. The intraepidermal group includes pemphigus vulgaris (PV), pemphigus foliaceous (PF), paraneoplastic pemphigus (PNPP), and immunoglobulin A pemphigus (IAP). The subepidermal group includes bullous pemphigoid (BP), mucous membrane pemphigoid (MMP), pemphigoid gestationis (PG), linear immunoglobulin A disease/chronic bullous disease of childhood (LAD/CBDC), bullous systemic lupus erythematosus (BSLE), lichen planus pemphigoidis (LPP), epidermolysis bullosa acquisita (EBA), and dermatitis herpetiformis (DH).

Studies that were undertaken worldwide on the epidemiology of AIBDs showed a great variation among populations [3]. There was a need to describe the epidemiological profile of AIBDs in Sudan because data was lacking. The objectives of our study were: (a) to define the spectrum of AIBDs in Sudan with their relative frequencies, (b) to determine the main clinico-epidemiological features for each AIBD, and (c) to estimate the burden of these diseases in Sudan.

## Methods

This was a retrospective cross-sectional study. We included all patients who were admitted to Khartoum Dermatological and Venereal Diseases Teaching Hospital (KDVTH). KDVTH is the main dermatology hospital in Sudan. In order to address the rarity of these diseases, we extended our coverage to 16 years from 2001 to 2016.

We used routinely collected health data in archived patients' records as our source of data. We extracted the following information from the patients' records: type of AIBD, age, gender, co-morbidities and onset (see S1 File for the data extraction tool). We included all patients with AIBDs between 2000 and 2016 and excluded records where the final diagnosis of the specific type of AIBD was not written. Data was managed using the Statistical Package for Social Sciences [4]. Descriptive analysis was used for clinico-epidemiological characteristics.

The study was approved by the ethical committees of the University of Khartoum and Khartoum State Ministry of Health. Consent was not obtained because according to the local ethical regulations, consent is not required when research uses routinely collected healthcare data. Confidentiality and anonymity were ensured throughout the work.

## Results

We identified 923 patients who had a final diagnosis of an AIBD. They represented 19.5% of the total 4736 patients who were admitted to the KDVTH between 2000 and 2016. The estimated annual rate of AIBDs at KDVTH was 57.7 cases per year which represented 1.3 per 100,000 population per year. For further analysis of the characteristics of each type of AIBD, we excluded patients where a final diagnosis was missing. The final number of included

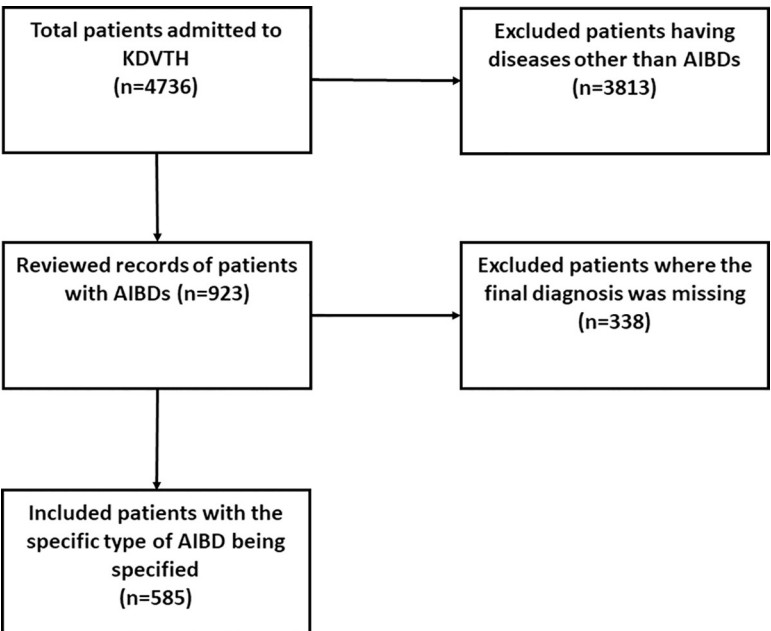

**Fig 1. Patient selection flow chart.** The inclusion of patient was a multi-step process. First, we identified patients with autoimmune bullous diseases (AIBD) from the total of patients in Khartoum Dermatological and Venereal Teaching Hospital (KDVTH) during the study period. Second, only patients with the specific diagnosis of an AIBD were included.

patients was 585 (Fig 1). Among the included patients, the male to female ratio was 1:1.6 (Fig 2). The cohort aged between two and 100 years, with a mean age of 45.2±22.1 years.

We found that only 52.7% of the patients went to the hospital in the first time the disease appeared, while 47.3% of the patients went to the hospital for a recurrent episode. In either

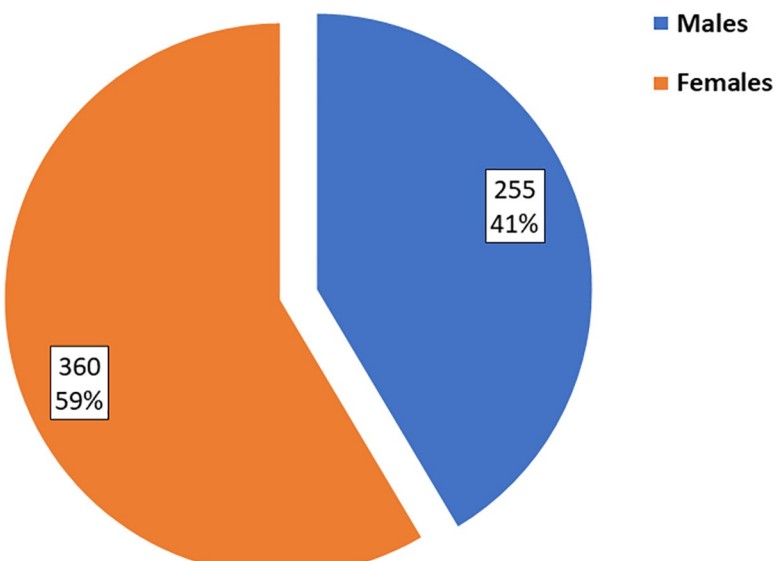

**Fig 2. Pie chart for gender distribution of patients included in the study.** Inclusion criteria was a final diagnosis of an autoimmune bullous diseases regardless of age or gender.

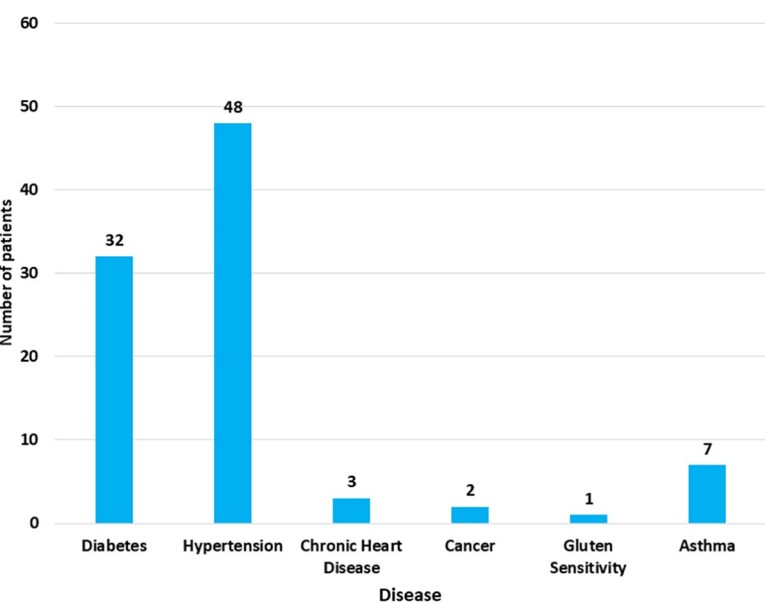

**Fig 3. Bar chart for the additional diseases in patients with autoimmune bullous diseases.** The total number of patients who had an additional chronic disease was 93.

case, patients waited from two days to 34.3 weeks after lesions appeared before they went to the hospital; the median was 4.2 weeks. About 16% (93/585) of the patients had one chronic disease or more in addition to the AIBD; most of them (86%) presented with hypertension, diabetes or both (Fig 3).

The intraepidermal group of AIBDs included 3355 patients (60.7%), while the subepidermal group included 230 cases (39.3%). All but one of the AIBDs were observed in this study; none of the patients had EBA (Table 1). PV was the most common disease (50.9%), followed by BP

**Table 1. Characteristics of the included 585 patients with different types of AIBDs.**

| AIBD | Number of cases (%) | Mean age ± SD (in years) | Gender (%) | | M:F ratio |
|---|---|---|---|---|---|
| | | | **Male** | **Female** | |
| **PV** | 298 (50.9%) | 40.4 ± 15.3 | 89 (29.9) | 209 (70.1) | 1:2.3 |
| **PF** | 48 (8.2%) | 46.3 ± 18.0 | 15 (31.3) | 33 (68.8) | 1:2.2 |
| **PNPP** | 2 (0.3%) | 51.5 ± 13.4 | 2 (100) | 0 (0) | Both males |
| **IAP** | 7 (1.2%) | 33.3 ± 16.8 | 1 (14.3)) | 6 (85.7) | 1:6.0 |
| **BP** | 165 (28.2%) | 66.0 ± 15.8 | 89 (53.9) | 76 (46.1) | 1:0.9 |
| **MMP** | 1 (0.2%) | 6 | 0 (0) | 1 (100) | A female |
| **PG** | 8 (1.4%) | 29.5 ± 9.7 | 0 (0) | 8 (100) | All females |
| **LAD/CBDC** | 49 (8.4%) | 9.0 ± 11.8 | 25 (51.0) | 24 (49.0) | 1:1 |
| **LPP** | 1 (0.2%) | 16 | 0 (0) | 1 (100) | A female |
| **BSLE** | 1 (0.2%) | 38 | 0 (0) | 1 (100) | A female |
| **EBA** | 0 | NA | NA | NA | NA |
| **DH** | 5 (0.9%) | 49.2 ± 24.3 | 4 (80) | 1 (20) | 1:0.3 |
| **Total** | 585 (100%) | 45.2±22.1 | 225 (38.5) | 360 (61.5) | 1:1.6 |

AIBD, autoimmune bullous disease; BP, bullous pemphigoid; BSLE, bullous systemic lupus erythematosus; DH, dermatitis herpetiformis; EBA, epidermolysis bullosa acquisita; IAP, immunoglobulin A pemphigus; LAD/CBDC, linear immunoglobulin A disease/chronic bullous disease of the childhood; LPP, lichen planus pemphigoides; M:F, male to female ratio; MMP, mucous membrane pemphigoid; NA, not applicable; PF, pemphigus foliaceous; PG, pemphigoid gestationis; PNPP, paraneoplastic pemphigus; PV, pemphigus vulgaris; SD, standard deviation.

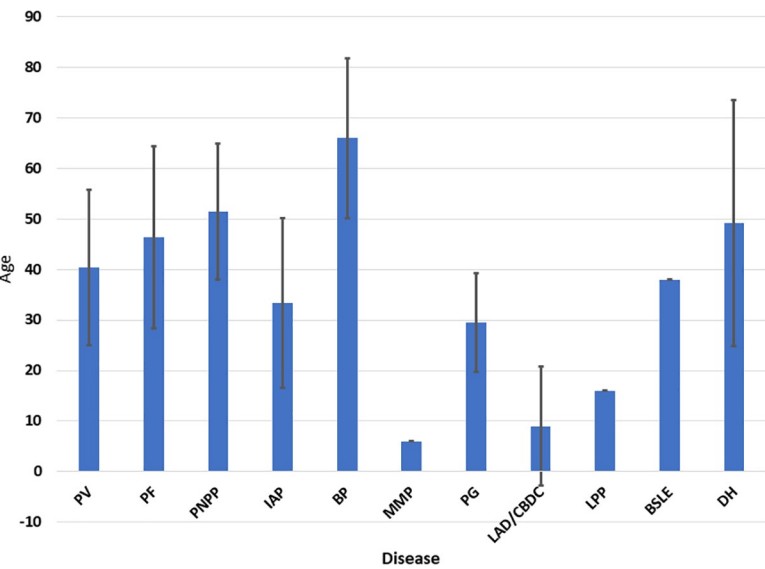

**Fig 4. Bar chart for the mean age at presentation and standard deviation for each autoimmune bullous disease.**
BP, bullous pemphigoid; BSLE, bullous systemic lupus erythematosus; DH, dermatitis herpetiformis; IAP, immunoglobulin A pemphigus; LAD/CBDC, linear immunoglobulin A disease/chronic bullous disease of the childhood; LPP, lichen planus pemphigoides; MMP, mucous membrane pemphigoid; PF, pemphigus foliaceous; PG, pemphigoid gestationis; PNPP, paraneoplastic pemphigus; PV, pemphigus vulgaris.

(28.2%), LAD/CBDC (8.4%), and PF (8.2%). PG and IAP constituted 1.4% and 1.2% of the cases, respectively. PNPP, MMP, LPP, BSLE, and DH were infrequent. Both PV and PF had a mean age of presentation in the fifth decade and occurred more commonly in females. In contrast, the mean age of BP was two decades later and affected males slightly more commonly than females. LAD/CBDC had a mean age of presentation in the first decade and affected both genders equally (Fig 4 and Table 1).

## Discussion

This is the first study in Sudan to investigate the AIBDs with description of the clinico-epidemiological characteristics of the whole spectrum; it included cases that were reported between 2000 and 2016. To the best of our knowledge, only seven studies worldwide have described the epidemiology of the whole spectrum of AIBDs in: Iran [5], Tunisia [6], Romania [7], Germany [8], Kuwait [9], China [10], and Malaysia [11]. S1 Table shows summarized findings of the studies that investigated the whole spectrum of AIBDs in different countries including our study. Most studies focused on an individual disease or a group of AIBDs. We were limited by using routinely collected health data that were not collected to answer the specific research questions, and hence there were missing data. We excluded records of patients where the final diagnosis was not written from the analysis of individual diseases. Apart from the literature-based study in China [10], our study covered the longest chronology of any empirical study on AIBDs and hence its findings are less prone to time fluctuations. In addition, KDVTH is the focus for dermatology in Sudan receiving daily hundreds of patients with skin diseases who come from all over the country by direct access to the hospital and by referrals from the governmental health system as well as private physicians.

This study highlights the high burden of AIBDs in Sudan. The rate of AIBDs at KDVTH that we found was higher than what was reported in Malaysia [11], China [10], Kuwait [9], Germany [8], Romania [7], and Tunisia [6]; it was only lower than the findings in Iran [5]

which is exceptionally characterized by a high prevalence of PV. Furthermore, the proportion of patients with AIBDs from the total patients in the hospital (19.5%) was higher than what was reported in Brazil and South Africa [12, 13]. Patient took long time before they sought treatment for their bullae, and many of them waited until lesions healed and reappeared in a subsequent episode. The Sudanese have been known for their weak treatment seeking behaviours [14]. People usually have an exceptionally low income, and services are centralized in the capital city which adds the costs of travelling.

We found that PV, BP, LAD/CBDC, and PF are the most common AIBDs in Sudan. PV in Sudan outnumbered BP [15], similar to Middle Eastern countries including Turkey [16], Malaysia [11], Kuwait [9], Tunisia [6], and Iran [5] in addition to Romania [7]. On the other hand, PV was less frequent than BP in European countries including Germany [8], Switzerland [17], and the United Kingdom [18] in addition to Singapore [19, 20] and Israel [21, 22]. The proportion of LAD/CBDC among AIBDs was relatively high in Sudan similar to Malaysia [11]. However, LAD/CBDC was uncommon in China [10], Kuwait [9], Germany [8], Romania [7], Tunisia [6], and Iran [5].

## The intraepidermal group

Our finding that PV was more common than PF is supported by reports from most countries including Tunisia [6], Saudi Arabia [23], Turkey [16], Iran [5], India [24], Japan [25], France [26], Italy [27], Singapore [19], and Israel [22]. However, PF was found more frequently than PV in Brazil [28] and other South American countries [29] where PF is endemic. PF was also more common than PV in some African countries, namely some parts of Tunisia [26], South Africa [30], and Mali [31].

Regarding gender distribution, PV followed the general tendency to occur in females more than in males, similar to most previous studies [5–7, 16–18, 22, 25, 27]. However, in some Arab Gulf countries, namely Kuwait [9] and Saudi Arabia [23], the disease was reported in males more than females. PF was also more common in females, a finding that corresponds to previous studies from Tunisia [6, 26], Kuwait [9], Iran [5], Japan [25], Brazil [28], South Africa [30], and Mali [31]. In contrast, more males were affected by PF than females in France [26], Turkey [16], Singapore [19], and Israel [22].

Regarding the age of presentation, PV had an average age at presentation in the fifth decade which is in good agreement with previous findings in Sudan [32], Turkey [16], Saudi Arabia [23], Iran [5], South Africa [30], and Singapore[19]. However, PV had a higher mean age at presentation in Japan [25] and the European countries: Germany [8], Romania [7], Switzerland [17], the United Kingdom [18], and Italy [27]. Like PV, PF had a mean age at presentation in the fifth decade in this study. Studies in South Africa [30], Mali [31], China [10], and Iran [5] were in line with the present study. In Italy [27], Singapore [19], Turkey [16], and Japan [25], it had a higher mean age than in the present study. Generally, the younger onset of these diseases in Sudan may be linked to the earlier accumulation of predisposing genetic factors with inducing local environmental factors. For instance, PV has been reported to be induced by insecticides [33], and there is extensive misuse of insecticides in Sudan [34].

The proportion of IAP among AIBDs was higher in this study than all previous studies that investigated the whole spectrum of AIBDs [5–11]. The percentage of IAP among intraepidermal AIBDs was higher in Sudan than Israel [22]. Females were affected with IAP more than males in the present study; however, in Japan [35] no gender predilection was noted, and in contrast, reports from Iran [5] showed that males were affected by IAP more than females. The mean age for IAP in the present study is very close to the findings in Kuwait [9], higher than the report from Iran [5], and lower than the results from Japan [35].

In Sudan, PNPP was extremely rare. This agrees with data from Malaysia [11], China [10], Kuwait [9], Germany [8], Romania [7], Tunisia [6], Iran [5], Turkey [16], and Singapore [19]. However, PNPP was more common in Japan [25]. This could be explained by the short survival time of patients with cancer in Sudan [36] which does not provide enough time for PNPP to develop.

## The subepidermal group

In Sudan, we found that BP was the most frequent subepidermal AIBD, and this is in agreement with all previous studies [5–11, 17, 20, 26, 37–39]. BP showed a slight male predilection in this study similar to the findings in Tunisia [6] and China [10]. However, females were affected more than males by BP in: Malaysia [11], Kuwait [9], Germany [8], Romania [7], Iran [5], Switzerland [17], Turkey [16], the United Kingdom [18], Singapore [20], Israel [21], Germany [37], France [39], the United States [40], and Uganda [38]. The mean age for BP was in the seventh decade similar to what was observed in Kuwait [9], Germany [8], Tunisia [6], and Turkey [16]. BP had a lower mean age in Iran [5] and Uganda [38] than in the present study. However, studies in Western countries including the United Kingdom [18], Switzerland [17], France [39], Germany [37], and the United States [40] in addition to Singapore [20] and Israel [21] reported a higher mean age.

The proportion of LAD/CBDC within the subepidermal group in the present study is similar to the findings in Uganda [38], but it is higher than the findings in Singapore [20], Germany [37], and France [39]. LAD/CBDC affected both genders equally in this study, similar to studies from China [10], Singapore [20], Germany [37], and Uganda [38]. However, there was a female predilection in Malaysia [11], Tunisia [6], and France [39], while more males than females were affected in Kuwait [9], Iran [5], and Denmark [41]. LAD/CBDC varied considerably among reported studies in the mean age of presentation. In the present study, predominance of the disease in the first decade was observed, similar to the finding in Germany [37]. Studies in Kuwait [9], Tunisia [6], and Uganda [38] showed dominant occurrence in the second decade. In Iran [5], Germany [8], Singapore [20], France [39], and Denmark [41], the mean age was even higher. The earlier presentation of the disease we observed in Sudan suggested that the frequency of the early-onset LAD/CBDC outnumbered that of the late-onset variant.

PG in Sudan was found less commonly than in Kuwait [9], Tunisia [6], and Germany [8]. This might be due to the mild nature of PG and so patients are less likely to be treated in KVDTH. However, studies in Romania [7] and Iran [5] found that PG occurred less frequently than in the present study, whereas none of the patients had PG in the studies from Malaysia [11], China [10], Singapore [20], and Uganda [38]. The mean age at presentation for PG was in the late twenties or early thirties in Kuwait [9], Germany [8], Tunisia [6], Iran [5, 42], and Saudi Arabia [43]; the present study confirmed this observation.

In Sudan DH was infrequent, similar to the results from Malaysia [11], China [10], and Romania [7]. However, it was found more commonly in Tunisia [6] and Germany [8]. DH showed a tendency to occur in males and in the fifth decade in the present study, a finding similar to that reported in Germany [8, 44]. However, in DH affected females twice as frequently as males and had a mean age of presentation in the third decade in Tunisia [6].

Only one case of MMP was reported in the present study. Previous findings which reported the rarity of MMP where in Malaysia [11], China [10], Kuwait [9], Romania [7], Tunisia [6], Iran [5], and Singapore [20]. This is possibly because patients with this disease may have been treated in other clinical facilities such as the dental or ophthalmology hospital. In contrast, MMP was relatively more common in Germany [8, 37] and Uganda [38]. Similarly, BSLE and

LPP were each found in one patient in the present study. This observation concurs with data indicating the rarity of these diseases from Malaysia [11], China [10], Kuwait [9], Germany [8], Romania [7], Tunisia [6], Iran [5], and Singapore [20]. However, LPP was slightly more frequent in Kuwait [9]. None of our cohort had EBA although it was reported to occur in China [10], Kuwait [9], Germany [8], Romania [7], Tunisia [6], Iran [5], Singapore [20], and Uganda [38].

### The way forward

Our study provided epidemiological data on the whole spectrum of AIBDs in Sudan. It guides the planning of health services and raises questions to be investigated. The large number of patients with AIBDs in Sudan necessitates future clinical trials and facilitates them at the same time [45]. The high burden and the earlier onset of AIBDs have raised questions on the genetic and environmental factors pertinent to the Sudanese for further investigation; this would provide precision medicine options for the Sudanese patients. Last but not the least, we recommend the establishment of a national registry for AIBDs in order to build a network for monitoring patients' needs all over the country and for responding promptly if health problems arise. In the time of COVID-19, the need for a registry is more crucial than ever before because it has been suggested that the immunomodulatory medications that patients with AIBDs receive may influence their risk of COVID-19 and hospitalization [46].

### Conclusions

Our research has led us to conclude that AIBDs represent an undeniable health problem to the Sudanese. Not only was the number of cases per year in KDVTH higher than in other studies, but it also represented a higher percentage of the total number of patients in the hospital. PV was the most common AIBD, followed by BP, LAD/CBDC, and PF. There was a considerable variation in the age and gender at presentation among patients with different types of AIBDs. Collectively, AIBDs affected females more commonly than males. Sudanese patients tended in general to present with AIBDs at an earlier age than what was noted in different parts of the world.

### Supporting information

**S1 File. Data extraction tool.**
(PDF)

**S1 Table. Summary of the studies that investigated the whole spectrum of AIBDs including our study.**
(PDF)

### Acknowledgments

We acknowledge the staff in the archive department of KDVTH for facilitating access to records, and Rofaida Elshafie for her assistance in data extraction.

### Author Contributions

**Conceptualization:** Omayma Siddig, Ahmed M. Suleiman.

**Formal analysis:** Omayma Siddig.

**Investigation:** Omayma Siddig.

**Methodology:** Omayma Siddig, Mayson B. Mustafa, Ahmed M. Suleiman.

**Project administration:** Omayma Siddig.

**Resources:** Omayma Siddig, Yousif Kordofani.

**Supervision:** Ahmed M. Suleiman.

**Visualization:** Mayson B. Mustafa, Yousif Kordofani, John Gibson, Ahmed M. Suleiman.

**Writing – original draft:** Omayma Siddig.

**Writing – review & editing:** Omayma Siddig, Mayson B. Mustafa, Yousif Kordofani, John Gibson, Ahmed M. Suleiman.

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
