## [Decision Letter · Decision Letter 0]

14 Dec 2020

PONE-D-20-35931

The epidemiology of autoimmune bullous diseases in Sudan between 2000 and 2016

PLOS ONE

Dear Dr. Siddig,

Thank you for submitting your manuscript to PLOS ONE. After careful consideration, we feel that it has merit but does not fully meet PLOS ONE’s publication criteria as it currently stands. Therefore, we invite you to submit a revised version of the manuscript that addresses the points raised during the review process.

We look forward to receiving your revised manuscript.

Kind regards,

Amit Sapra

Academic Editor

PLOS ONE

Journal Requirements:

2) Please include additional information regarding the data extraction tool used in the study and ensure that you have provided sufficient details that others could replicate the analyses. For instance, if you developed a data extraction tool as part of this study and it is not under a copyright more restrictive than CC-BY, please include a copy, in both the original language and English, as Supporting Information, or include a citation if it has been published previously.

3) Please use standard statistical reporting for your estimates. For example, rates can be presented as 57.7 per 100,000 population and so forth.

4) We note that you have indicated that data from this study are available upon request. PLOS only allows data to be available upon request if there are legal or ethical restrictions on sharing data publicly. For information on unacceptable data access restrictions, please see http://journals.plos.org/plosone/s/data-availability#loc-unacceptable-data-access-restrictions.

Reviewers' comments:

Reviewer's Responses to Questions

**Comments to the Author**

1. Is the manuscript technically sound, and do the data support the conclusions?

Reviewer #1: Yes

2. Has the statistical analysis been performed appropriately and rigorously? 

Reviewer #1: Yes

3. Have the authors made all data underlying the findings in their manuscript fully available?

Reviewer #1: Yes

4. Is the manuscript presented in an intelligible fashion and written in standard English?

Reviewer #1: Yes

5. Review Comments to the Author

Reviewer #1: 1) the study design should be- Retrospective Cross Sectional Study. 2) Lack of consent - can present as an ethical consideration 3) Grammatical errors which need correction 4) Better to mention the percentages of males and females being affected for each AIBD group should be mentioned rather than Male: Female ratio. 5) Bar graph representation of the age of patients being affected for each AIBD group can be included for a better visual representation. 6) Authors have included a Fig 3 ( bar graph) showing various comorbidities that the patients had. I think it would be better if the authors elaborate on the same. For an example if they found a higher prevalence of an AIBD's with a particular underlying comorbidity, that would be very informative.

6. PLOS authors have the option to publish the peer review history of their article (what does this mean?). If published, this will include your full peer review and any attached files.

Reviewer #1: No

---

## [Author Response · Author response to Decision Letter 0]

20 Jan 2021

1) We double checked that the manuscript meets PLOS ONE's style requirements.

2) We included the data extraction tool as Supporting Information, named S1_File. We re-uploaded the previous S1_File.pdf as S2_File. We developed the tool in English and therefore we are uploading it as it is.

3) Standard reporting of the rates was used (lines 25, 74).

4) Regarding data availability, the ethics committees of the University of Khartoum and Khartoum State Ministry of Health restricted publicly sharing of research data that were extracted from routinely-collected healthcare data even if anonymized. Some of the diseases under study are very rare and therefore potentially identifying. Data are available from the University of Khartoum Faculty of Dentistry (dentistry@uofk.edu).

Reviewer’s comments:

1) The study design is now written as advised (lines 18 and 57).

2) We agree that a lack of consent is an ethical consideration in general. However, for a retrospective study using routinely-collected health care data, this is a grey area. The ethics committees exempted this study from the requirement of informed consent based on its minimal risk to ethics.

3) Grammatical errors were corrected in lines: 19, 24, 35, 50,64, 65, 78, 123, 132, 188, 212, and 240.

4) We added information on gender percentages for each disease to Table 1. We still believe that the male-to female ratio is important for comparison with other countries.

5) A bar chart for age representation is uploaded as Fig4 and appropriately cited within the manuscript.

6) We did not observe a high prevalence of a co-morbidity with a specific AIBDs.

---

## [Decision Letter · Decision Letter 1]

22 Mar 2021

PONE-D-20-35931R1

The epidemiology of autoimmune bullous diseases in Sudan between 2000 and 2016

PLOS ONE

Dear Dr. Siddig,

Thank you for submitting your revised manuscript to PLOS ONE. After careful consideration, we feel that it has merit but does not fully meet PLOS ONE’s publication criteria as it currently stands. Therefore, we invite you to submit a revised version of the manuscript that addresses the points raised during the review process.

We look forward to receiving your revised manuscript.

Kind regards,

Feroze Kaliyadan, M.D.

Academic Editor

PLOS ONE

Journal Requirements:

Reviewers' comments:

Reviewer #2: I appreciate the authors attempt to evaluate ‘The epidemiology of autoimmune bullous diseases in Sudan between 2000 and 2016’. Some minor comments

1. “However, in some Arab Gulf countries, namely Kuwait [9] and Saudi Arabia [23], the disease was reported in males more than females suggesting an element of gender inequality in those countries [32]….” (Lines 151-153). Strongly recommend to delete the wording ‘gender inequality in those countries’. How will a difference in male:female ratio in a country denote an element of gender inequality? The reference 32 as cited by the authors was commenting on concluded that “his study highlighted important differences in access to and utilisation of PHCS between urban and rural populations in Riyadh province in the KSA. These findings have implications for policy and planning of PHCCs and reducing inequalities in health care between rural and urban populations…” Hence the authors claim is neither supported by the cited literature. There are AI conditions where there may be male:female incidence differences. In the same paragraph, authors were commenting “In contrast, more males were affected by PF than females in France [26], Turkey [16], Singapore [19], and Israel [22]…” Whether authors inference that there is ‘an element of gender inequality’ in these countries too! Moreover, in Line 184, authors were stating there is higher female preponderance for BP in Kuwait.

2. The authors rightly mentioned in the manuscript regarding retrospective study design and possibility of missing the information (as could be extracted from a leading question). Please highlight specifically the lack of information on neurological disease a/w AIBD in this study.

---

## [Author Response · Author response to Decision Letter 1]

29 Jun 2021

Omayma Siddig

Faculty of Dentistry, University of Khartoum

Khartoum 11111

Sudan

29 June, 2021

Ref.: PONE-D-20-35931, The epidemiology of autoimmune bullous diseases in Sudan between 2000 and 2016, second revision

Dear Feroze Kaliyadan

We thank you for the flexibility you showed regarding the due date for submission this manuscript. We also thank the reviewer for their time reviewing and commenting on the manuscript.

We have edited the manuscript as requested and here are the details of the changes made:

1) Regarding reference list: we removed the reference related to the reviewer’s first comment, and we added a reference reporting on the outcomes of COVID-19 in patients with autoimmune bullous as we believe it enforces the idea of a national registry.

2) We uploaded the pictures that have been corrected by the PACE tool as advised. 

Reviewer’s comments:

1) We deleted the phrase “suggesting an element of gender inequality in those countries”. The reviewer made a valid point and we agree with them.

2) Although the association between neurological diseases and autoimmune bullous diseases is an interesting area and yes our study lacked any information about neurological diseases, we cannot comment on this issue. We feel that it would be biased to comment on association between neurological diseases and autoimmune bullous diseases only and leave all other associations that have been reported in the literature as if we imply that there is definitely an association. On the other hand, we cannot comment on every single association that was reported as it would be overwhelming for the reader and would put the manuscript off track.

Attached along with this letter are the marked-up manuscript with track changes, the unmarked manuscript, the PACE-corrected figures and the supplementary files.

We look forward to receiving from you the acceptance of publication in PLOS ONE. We would be happy to respond to any questions or comments that may arise. 

Kind regards,

Omayma Siddig; BDS, MFD, MFDS, MSc

Faculty of Dentistry, University of Khartoum

On behalf of all authors.

---

## [Editor Report · Decision Letter 2]

1 Jul 2021

The epidemiology of autoimmune bullous diseases in Sudan between 2000 and 2016

PONE-D-20-35931R2

Dear Dr. Siddig,

We’re pleased to inform you that your manuscript has been judged scientifically suitable for publication and will be formally accepted for publication once it meets all outstanding technical requirements.

Kind regards,

Feroze Kaliyadan, M.D.

Academic Editor

PLOS ONE

---

## [Editor Report · Acceptance letter]

5 Jul 2021

PONE-D-20-35931R2 

The epidemiology of autoimmune bullous diseases in Sudan between 2000 and 2016 

Dear Dr. Siddig:

I'm pleased to inform you that your manuscript has been deemed suitable for publication in PLOS ONE. Congratulations! Your manuscript is now with our production department. 

Kind regards, 

on behalf of

Dr. Feroze Kaliyadan 

Academic Editor

PLOS ONE